# How Does the Urban Built Environment Affect Online Car-Hailing Ridership Intensity among Different Scales?

**DOI:** 10.3390/ijerph19095325

**Published:** 2022-04-27

**Authors:** Guanwei Zhao, Zhitao Li, Yuzhen Shang, Muzhuang Yang

**Affiliations:** 1School of Geography and Remote Sensing, Guangzhou University, Guangzhou 510006, China; zhaogw@gzhu.edu.cn (G.Z.); 2112001071@e.gzhu.edu.cn (Z.L.); 2112001045@e.gzhu.edu.cn (Y.S.); 2Institute of Land Resources and Coastal Zone, Guangzhou University, Guangzhou 510006, China

**Keywords:** urban built environment, online car-hailing, multiscale, spatial nonstationary

## Abstract

Understanding the effect of the urban built environment on online car-hailing ridership is crucial to urban planning. However, how the effects change with the analysis scales are still noteworthy. Therefore, a multiscale exploratory study was conducted in Chengdu, China, by using the stepwise regression selection and three spatial regression models. The main findings are summarized as follows. First, as the grid size increases, the number of built environment factors that have significant effects on trip intensity decrease continuously. Second, the effects of population density and road density are always positive from the 500 m grid to the 3000 m grid. As the analysis scale increases, the effect of proximity to public transportation shifts from inhibitory to facilitation, while the positive effect of land-use mix becomes stronger. Land-use type has both positive and negative effects and shows different characteristics at different scales. Third, the effects of built environment factors on online car-hailing trip intensity show different spatial variability characteristics at different scales. The effect of population density gradually decreases from north to south. The effect of road network density shows circling and wave patterns, with the former at relatively fine scales and the latter at relatively coarse scales. The spatial variation in the effect of land-use mix can only be observed more significantly at a relatively coarse scale. The effect of bus stop density is only obvious at the relatively fine and medium scales and shows a wave-like pattern and a circle-like pattern. The effect of various land-use types shows different spatial patterns at different scales, including wave-like pattern, circle-like pattern, and multi-core-like pattern. The spatial variation in the effects of various land-use factors gradually decrease with the increase in the analysis scale.

## 1. Introduction

Before the advent of online car-hailing (e.g., Uber/Lyft/Didi), taxis were one of the important means of transportation for urban residents in their daily travel, due to providing flexibility and personalized service capabilities in the current transportation system [1]. However, the cruise-style service of traditional taxis cannot quickly and efficiently match travel demand and supply in time and space. With the development of location-based services and mobile internet technologies, online car-hailing which can efficiently integrate travel supply and demand information has brought a huge impact to traditional taxis [2]. For example, traditional taxis can take passengers only by street hailing, while online car-hailing carries passengers through the combination of network appointments and street hailing. Due to the personality, convenience, and flexibility, online car-hailing plays an increasingly important role in people’s daily travel activities [3]. The company Didi is one of leading mobile transportation platforms in China, which offers diversified transportation services to 450 million users in more than 400 cities [4]. A large amount of trajectory data were produced during the running of online-hailing vehicles. These data provide a good foundation for studying the temporal and spatial laws of residents’ travel, discovering travel hotspots, revealing the urban spatial structure, identifying the functional areas of the city, etc.

As an area with a high concentration of resident activities, the proper planning of urban transportation functions is an important factor in the sustainable development of a city, as well as for the efficiency of people’s work and their well-being. In addition, air pollution from the transportation industry has had a serious impact on urban environmental problems, especially in developing countries with rapid urbanization. As one of the main ways of green travel, online car-hailing can effectively reduce the carbon emission of residents’ travel and improve the quality of the urban environment. Therefore, understanding the spatiotemporal characteristics of online car travel and its relationship with the built environment can not only prompt the transportation infrastructure design, but also provides a basis for guiding low-carbon transportation behavior.

The urban built environment, which is the human-made environment provided for human activities, usually includes the spatial environment formed by the interaction of multiple factors such as land-use, transportation infrastructure, and urban design [5]. In fact, the rapid development of the online car-hailing transportation mode will also bring corresponding difficult challenges to urban traffic planning and management [6,7], especially on how to reasonably integrate the built environment policies (e.g., regional development planning, comprehensive land development, and street network construction) with transportation policies (e.g., relevant policies such as bus, taxi, and online car-hailing operation management) [7]. In order to integrate the built environment policies with transportation policies, the effect of the built environment on online car-hailing needs to be explored comprehensively. Previous studies have shown that the relationship between urban transportation and the built environment is significant and complicated [8,9]. For example, Cervero et al. found that the built environment, with high density and a high degree of mixing and grid street network, has a significant impact on reducing travel distance and encouraging nonmotorized travel [10]. Ge et al. found that the health care area is the most critical factor in all land-use variables that impact taxi ridership [11]. Yang et al. found a positive correlation between accessibility to subways and taxi ridership [12]. However, to the best of the authors’ knowledge, more efforts should be made to analyze the effect of the built environment on online car-hailing travel, especially the multiscale effect; that is, how do urban built environment factors affect online car-hailing ridership intensity among different scales? Answering this question can help provide a basis for decision-making in the planning of transport policies.

Our study attempts to answer this question from three aspects. First, what are the differences in built environment factors that have significant effects on online car-hailing travel behavior between different scales? Second, how do such factors affect online car-hailing travel behavior in different scales? Third, how do the effects of these built environment factors vary in the space of different grids? Therefore, this study focuses on three main tasks. Firstly, based on the principle of 5Ds developed by Ewing and Cervero [13], we apply the stepwise regression method to improve the selection of independent variables of the urban built environment in ten grid scales. Secondly, we construct the ordinary least square regression models from global perspectives of spatial relationship, and then analyze the impact of built environment factors on the online car-hailing ridership intensity. Finally, we explore the spatial varieties of the effect of built environment factors on the online car-hailing ridership intensity based on the local regression results derived by the MGWR model.

The structure of this paper is organized as follows. In Section 2, we review related research on the effect of the urban built environment on car-hailing travel behavior. In Section 3, we describe the study area, data, and methods including stepwise regression, Moran’s I test, ordinary least square regression model, geographically weighted regression model, and multiscale geographically weighted regression model. In Section 4, we use several spatial regression models to fit the multiscale relationship between the urban built environment and online car-hailing ridership intensity in detail and analyze the regression results. In Section 5, we make the conclusions and note the limitations. The findings can provide invaluable insights into policy formation regarding online car-hailing to guide transport development in Chengdu, China, and worldwide.

## 2. Literature Review

Taxis used to be the main cruising car rental service in the city before online car-hailing [14]. Therefore, extensive research was conducted on many aspects using taxi data, such as spatiotemporal characteristic analysis of travel behaviors, transportation network recognition, traffic demand prediction, the impact of the built environment on taxi travel behavior, the relationship between taxis, buses and subway, and so on [2,9,11,12,15,16,17,18,19]. Numerous scholars believe that taxi trajectory data can reflect the temporal and spatial characteristics of urban traffic flow to a certain extent [20], and the spatiotemporal heterogeneity shown by it is closely related to the urban form. In recent years, an increasing number of researchers began to focus on the influence of the built environment on the travel behavior of car-hailing. For example, according to Jiang’s research in 2009, the human mobility pattern is mainly attributed to the underlying street network [21]. Li et al. identified the preferred pick-up places for passengers and drivers [22]. Cervero et al. found that the improvement in the degree of land-use mixture can reduce the vehicle miles traveled (VMT) [10]. A positive correlation between accessibility to subway and taxi ridership was found by Yang et al. [12]. Ge et al. claimed that among all land-use variables, the health care area has the most critical impact on taxi ridership [11]. Liu et al. explored the influence mechanism of different built environment variables on car ownership [23]. These studies have comprehensively explored the relationship between the built environment and the travel behavior of car-hailing. However, due to the difference between traditional taxi-hailing and online car-haling, the influence of the built environment on the online car-haling travel behavior is still a hot issue needing to be explored.

In terms of the “five Ds” principle, numerous studies have been facilitated to investigate the relationship between the built environment and the travel behavior of online car-hailing. For example, these scholars claimed that the effects of population density, road network design, and land-use mix on sharing traveling demand are mainly positively correlated [14,24,25,26]. Due to the floating feature of sharing travel, the studies on the impact of destination accessibility on online ride-hailing travel are relatively rare [27]. These scholars claimed that distance to transit, which represents the ease of access to transit services, can prompt an increase in ride-hailing demand [25,28]. This finding was consistent with Deka’s travel survey results, which claim that residents living near transit stations use ride-hailing more frequently [29]. In addition, Kong et al. [30] and Wang and Noland [31] also found a positive relationship between proximity to the metro station and online car-hailing trips.

Although these studies have provided clues for evaluating the effect of the built environment on the travel behavior of online car-hailing in our study, there are still two main problems that need to be answered. First, the measurement factors of the built environment were not fully controlled when selecting variables in the above study [6], which may lead to inconsistent and even contrary results. For example, Yu and Peng found that an increase in road density could generate more online car-hailing travel [24,25], while Li et al. [7] and Sabouri et al. [32] indicated that the road density had no appreciable impact or even a negative impact on the ridership of online car-hailing. Second, the scales of the analysis unit in previous studies were inconsistent, which also may lead to different conclusions. As we know, the analysis unit is an important issue in travel behavior studies. The traditional traffic analysis zone (TAZ) is the dominant analysis unit in the traffic behavior research [33,34,35]. The division types of TAZ usually include administrative division, census zones, land-use parcel, residential district, grid geometry, and so on. It is demonstrated that TAZ in the form of grid geometry with appropriate size will reflect the impact of the built environment on travel behavior more accurately [36]. Yet, it is also hard to determine an appropriate size of the grid geometry under different research scenarios [37]. Coined by geographers during the 1970s, the modifiable areal unit problem (MAUP) [38] is one of the most stubborn problems in geography analysis when spatially aggregated data are used. The modifiable areal unit problem in spatial statistical analysis usually means that analyzing the datasets in different analysis units (such as shape, scale) will likely provide inconsistent results [39]. The uncertainty caused by the MAUP impacts the robustness and reliability of statistical results. Some scholars have explored the MAUP issue in sharing transport. For example, Gao et al. studied the MAUP issue of dockless bike sharing usage [40]. However, the MAUP issue in online car-hailing behavior modeling results are limited. Therefore, the differences between the built environment’s effect on online car-hailing in different scales need to be explored.

In the term of research methods, the effect of built environment factors on car travel behavior is usually quantified by the ordinary least square (OLS) regression model [12,34,35], due to its efficiency and convenience. However, there are still some limitations in the OLS model, such as inconsistent parameters or inaccuracy of test results, as the spatial nonstationary aspect of online car-hailing travel is ignored by the OLS model [41,42]. Therefore, the spatial effect should be considered in the regression model. Geographically weighted regression is an analysis method that deals with the spatial nonstationary aspect of the relationship, which was proposed by Brunsdon, Fotheringham, and other scholars in 1996 [43]. At present, GWR and its extensions have developed into one of the important methods of spatial relationship analysis. Qian et al. analyzed the spatial variations in taxi behavior in New York City using the GWR model [17]. Zhang et al. investigated the relationship between the taxi traffic spatiotemporal congestion pattern and the built environment [44]. Wang et al. analyzed the local effects of land use and capital investment using the global model and GWR model [14], respectively, and in the local results there appeared a local spatially inverse relationship in the areas adjacent to the international airport, indicating the existence of negative externalities. However, the traditional GWR model often uses a single kernel function and a unified “best average” bandwidth to calculate the weights, which ignores the spatial scale difference in variable estimation. Compared with the traditional GWR model, the multiscale geographically weighted regression model can reflect the scale differences in variable estimates in the multivariate GWR model [45,46,47]. In the context of increasingly diverse data modeling scenarios, the scales of spatial data are becoming more and more complex [48]. Previous research demonstrated that multiscale GWR technology can achieve model estimation more accurately and comprehensively due to its robustness and universality [49]. Some existing work claimed that the multiscale GWR model should be the preferred technology for the multivariate GWR model [45,48,49,50]. However, the travel behavior of online car-hailing is different than the behavior of traditional taxis. To our best knowledge, comparative studies of various spatial models among different scales are relatively rare in the quantitative analysis of the impact of land use on online car-hailing travel. In short, the impact of the urban built environment on online car-hailing travel needs to be strengthened, especially for the spatial variations in effects in diverse scales.

To this end, this article tries to analyze how the built environment affects the online car-hailing travel intensity in different scales, through spatial regression models such as the OLS model, the GWR model, and the MGWR model by taking Chengdu, China, as the study area. The results of this paper can not only enrich the theoretical connotation and research method of transportation geography, but also provide a reference for online car-hailing management and transportation policy optimization.

## 3. Materials and Methods

### 3.1. Study Area

Chengdu city was selected as the study area of our research. Chengdu is the capital of Sichuan Province, which is an important central city in western China approved by the State Council, and a comprehensive transportation hub. Chengdu has 12 municipal districts, 3 counties, and 5 county-level cities under its jurisdiction, with a total area of 14,335 square kilometers. The main data of the seventh national census of Sichuan Province show that the permanent population of Chengdu reached 20.938 million by the end of 2020 [51]. Chengdu has a flat terrain, vertical and horizontal river networks, and accessible roads. It has a humid subtropical monsoon climate. According to statistics from the national online car-hailing regulatory information exchange platform in March 2021, the number of Chengdu’s online car-hailing vehicle licenses is the largest among the 36 central cities (municipalities, provincial capitals, and cities under separate planning) in the country, and the number of online car-hailing orders ranks second among 36 central cities [52]. It can be seen that online car-hailing has already occupied a pivotal position in the travel system of Chengdu residents. Since more than 98% of the trip origins and destinations fell within the 89 streets of Chengdu, the study area was set to coincide with the area where the 89 streets are located. The location of the study area is shown in Figure 1.

### 3.2. Data Source and Preprocessing

The online car-hailing order data were provided by the Didi Gaia open dataset project (https://gaia.didichuxing.com, accessed on 24 January 2021). The data period is from 7 November 2016 to 13 November 2016, with a total of 1,610,652 records. The weather conditions in Chengdu during this period were good and did not have a major impact on online car-hailing travel. The original data include fields such as order number (encrypted and desensitized), pick-up time, drop-off time, pick-up location longitude, pick-up location latitude, drop-off location longitude, and drop-off location latitude. The coordinate system of the longitude and latitude fields is GCJ-02, which is the official Chinese geodetic datum formulated by the Chinese State Bureau of Surveying and Mapping. The coordinates of pick-up and drop-off location were converted to the WGS1984 coordinate system. The administrative boundary vector data were collected from the road traffic monitoring platform of Chengdu; the collection time was June 2021, the data format is shapefile, and the coordinate system is the WGS1984 coordinate system. POI data were collected from Gaode Map (https://lbs.amap.com/, accessed on 7 March 2018), which includes fourteen POI categories: catering facilities; scenic spots; public service facilities; companies; shopping facilities; transportation facilities; financial facilities; educational, scientific, and cultural facilities; residence district; living service facilities; sports and leisure facilities; medical service facilities; government agencies; and accommodation service facilities. By cleaning the original POI data, 274,175 POI records were obtained for further analysis. Population density was calculated using the WorldPop dataset with a resolution of 100 m (https://www.worldpop.org/, accessed on 17 April 2020). The bus station data were extracted from the transportation facilities category of POI data. Finally, the coordinate system of all spatial data was unified as the project coordinate system of WGS1984 UTM Zone 48N.

Traffic analysis zone is the geography unit that is most commonly used in conventional transportation planning models. The type and spatial extent of zones typically varies in models. The square grid is one of the classic geography units for transportation analysis. There is no doubt that the gridded map with appropriate cell size can reveal the spatial varieties of transportation patterns more clearly. However, there is still no authoritative and uniform standard for obtaining grid size in academia. The purpose of this paper is to investigate how the built environment affects the intensity of online car trips in different grid scales. The criteria for scale division mainly includes two aspects. First, the minimum grid size for previous studies of online car-hailing in Chengdu is 500 m. Therefore, 500 m was set as the minimum analysis size for this paper. Second, as approximately 93% of the online taxi trip distances in this paper are shorter than 5000 m, 5000 m was set as the maximum analysis size in this paper. To ensure that we can reveal the characteristics of the effect of the built environment on online car-hailing trips at different scales, while reducing the computational burden, 500 m was chosen as the criteria for scale division. Then, we created ten grids with sizes ranging from 500 m to 5000 m at 500 m intervals, by using the create fishnet tool of ArcGIS software (Esri, Redlands). Moreover, we used the tools of ArcGIS software (Esri, Redlands) such as zoning statistics, spatial join, and field calculator to calculate the built environment indicator values for each cell. In addition, all data were standardized using zero-mean normalization method.

### 3.3. Methods

#### 3.3.1. Indicator System of Urban Built Environment

The well-known “five Ds” principle, which was proposed by Ewing and Cervero in 2010, is one of the most widely used metrics to measure the urban built environment. According to the “five Ds” principle, the urban built environment can be measured from five aspects: design, diversity, density, distance to transit, and destination accessibility. The destination accessibility is usually measured by distance to the CBD (central business district) in previous studies. However, the assumption of using distance to the CBD as an indicator is that the CBD includes the vast majority of destinations for taxi trips. The CBD in our study includes subdistricts such as Chunxilu, Hongxinglu, Yanshikou, Luomashi, and Shunchengjie, with a total area of about 2.45 km^2^. According to the drop-off distribution of our study, the CBD does not hold the majority of destinations for taxi trips. In addition, the main purpose of our study is to explore how the urban built environment affects online car-hailing trip (pick-up) intensity among different analysis scales. Previous studies show that land use has a great impact on the trip generation. Therefore, we chose different land-use types to replace the destination accessibility. Eighteen indicators were selected as the initial variables following the modified “five Ds” principle (see Table 1).

The diversity was measured by the land-use mix based on the principle of the Herfindahl Hirschman Index (HHI), which is a classic metric to reflect diversity [53]. The formula of land-use mix is shown as follows:(1)LMi=∑j=1k(NijNi)2
where $LM_{i}$ denotes the land-use mix of cell i, $N_{i}$ is the total POI numbers in cell i, $N_{ij}$ is the total POI numbers of type j in cell i, and k is the category of POIs. For ease of description, a simplified name is given for each metric (see Table 2).

#### 3.3.2. Independent Variable Selection Based on Stepwise Regression

As mentioned above, eighteen initial indicators were selected to represent the urban built environment based on the “5Ds” theory. However, the multicollinearity test was not performed. It is well known that multicollinearity is a common problem when two or more of the predictors in a regression model are moderately or highly correlated. Variance inflation factors (VIF) is a commonly used metric to help detect multicollinearity. The VIF values of indicators in different scales are shown in Table 3.

Using the VIF value greater than ten as the criterion for the multicollinearity test, it can be seen from Table 3 that there are multicollinearity problems in the selected independent variables at all scales. Since the multicollinearity problem can seriously affect the performance of the regression model, it must be dealt with. Stepwise regression is a classic method for explanatory variable selection, which mainly solves the problem of multicollinearity among variables. The commonly used stepwise regression methods are forward selection, backward elimination, and bidirectional elimination. In this paper, we use bidirectional elimination for the selection of optimal independent variables. The bidirectional elimination is a combination of the forward selection and backward elimination, testing at each step for variables to be included or excluded. In general, the process of bidirectional elimination consists of two basic steps: one is to remove variables from the regression model that are not significant by t-tests, and the other is to introduce new variables into the regression model that are significant by F-tests. The bidirectional elimination algorithm in this paper is implemented programmatically using the python language and the statistical analysis software named statsmodels. The detailed information of statsmodels can be found at the URL: https://www.statsmodels.org/stable/index.html (accessed on 15 June 2021).

#### 3.3.3. Ordinary Least Squares (OLS) Regression

Regression analysis is a quantitative technique that studies the relationship between the dependent variable and the independent variable. OLS regression is a regression analysis method that uses ordinary least squares to model the linear relationship between the independent variable and the dependent variable. The OLS model is usually the starting point of spatial regression analysis. The common form of the OLS regression equation is as follows:(2)$$y_{i}=\sum_{i=1}^{n}\beta_{i}x_{i}+\varepsilon_{i}$$

In Formula (2), $y_{i}$ is the value of the dependent variable; $x_{i}\;\left(i=1,2,\ldots n\right)$ is the value of the independent variable; $\beta_{i}\left(i=1,2\ldots n\right)\;$ is the coefficient of regression model; $\varepsilon_{i}$ is the error term of the model.

The OLS model is a global regression model, which is usually used to identify the significant built environment variables. However, the value of the regression coefficient estimated by the OLS model is the average value of the entire study area, which cannot reflect the spatial variation in the regression parameters. Therefore, the spatial variation in the coefficients needs to be addressed using local regression techniques such as the GWR series model.

#### 3.3.4. Moran’s I Test

Spatial nonstationarity is a prerequisite for the application of the GWR. The test for spatial nonstationarity is to measure the presence of spatial autocorrelation in the residual distribution of the regression model. Moran’s I is an important measure of global spatial autocorrelation proposed by Patrick Alfred Pierce Moran in 1950 [54]. The value range of the Moran’s I is between −1 and 1. A positive value of Moran’s I indicates that the data exhibit positive spatial autocorrelation, and the larger the value, the more obvious the spatial autocorrelation. A negative value of Moran’s I indicates that the data exhibit negative spatial autocorrelation, and the smaller the value, the greater the spatial variation. A zero value of Moran’s I indicates that the spatial distribution is random. The *p*-value and the Z-score are required to interpret the Moran’s I test result. That is, the Moran’s I test result is considered valid when the *p*-value is less than 0.05 (passing the 95% confidence test) and the Z-score exceeds the critical value of 1.65 (the threshold set by rejecting the null hypothesis). The calculation formula of Moran’s I is as following:(3)$$I=\frac{\sum_{i=1}^{n}\sum_{j=1}^{n}\omega_{i,j}z_{i}z_{j}/S_{0}}{\sum_{i=1}^{n}z_{i}^{2}/n}$$
where $\omega_{i,j}$ is spatial weight between sample i and sample j; n is the number of samples; $z_{i}$ ($z_{j}$) is the difference between sample i (j) and the mean of all samples; $S_{0}$ is the sum of all spatial weights.

#### 3.3.5. Geographically Weighted Regression

Spatial heterogeneity is a universal phenomenon in geography, which is also the premise of the existence of the first law of geography. Therefore, it is necessary to consider the spatial heterogeneity of variables when carrying out regression analysis for geographic phenomena. GWR is a locally weighted regression analysis model about location. GWR quantifies the heterogeneity or nonstationary characteristics in the spatial data relationship through the parameter estimation results that change with different locations. Because the local effects of spatial objects are taken into account, GWR has higher accuracy than OLS in the aspect of spatial regression. The basic GWR model can generally be expressed as follows:(4)$$y_{i}=\beta_{0}\left(u_{i},v_{i}\right)+\sum_{k=1}^{m}\beta_{k}\left(u_{i},v_{i}\right)x_{ik}+\varepsilon_{i}$$

In Formula (4), $y_{i}$ is the value of the dependent variable at position i; $x_{ik\;}\left(k=1,2,\ldots m\right)$ is the value of the independent variable at position i; $\left(u_{i},v_{i}\right)$ are the coordinates of position i; $\beta_{0}\left(u_{i},v_{i}\right)$ is the intercept term; $\beta_{k}\left(u_{i},v_{i}\right)\left(k=1,2,\ldots m\right)\;$ is the coefficient of regression model; $\varepsilon_{i}$ is the error term of the model.

#### 3.3.6. Multiscale Geographically Weighted Regression

Multiscale geographically weighted regression is an extension of traditional geographically weighted regression developed by Fotheringham et al. [47]. MGWR relaxes the assumption that all processes to be modeled are on the same spatial scale, and can be regarded as a geographically weighted regression model with spatially variable parameters. Previous studies have shown that the MGWR model that takes into account the bandwidth of variability has stronger explanatory power and robustness than the traditional GWR model [50,55]. The calculation formula of the MGWR model is as follows:(5)$$y_{i}=\beta_{0}\left(u_{i},v_{i}\right)+\sum_{k=1}^{m}\beta_{bwk}\left(u_{i},v_{i}\right)x_{ik}+\varepsilon_{i}$$

In Formula (5), bwk represents the bandwidth used by the regression coefficient of variable k. The meanings of the remaining variables are the same as variables in Formula (4).

#### 3.3.7. Model Evaluation Metrics

Three commonly used metrics, determination coefficient (R^2^), residual sum of squares (RSS), and Akaike information criterion (AIC) [56], were selected to evaluate the model performance. R^2^ is mainly used to evaluate the goodness of fit of the model. The higher the R^2^ value, the better the fitting performance of the model. RSS is mainly used to measure the deviation between the measured value and the predicted value of the dependent variable. The lower the RSS value, the closer the model’s estimated result is to the actual measured value. AIC can be used to measure the practicality and complexity of the model. The smaller the AIC value, the better the fitting effect of the model.

## 4. Results and Discussion

### 4.1. The Global Regression Results Using the OLS Model in Different Scales

The global regression results of the OLS model using the stepwise method and model evaluation information are shown in Table 4.

As can be seen from Table 4, the built environment factors that affect the intensity of online car-hailing trips vary among different scales. The *p*-values show that the involved variables are significant at least in the level of 0.05. The amount of built environment factors that affect the online car-hailing travel intensity decreased with the increasing of the analysis scale. The details are described for each scale in the following text.

At the scale of 500 the m grid, the positive influence of built environment factors, such as residential district density, accommodation service facility density, public service facility density, financial facility density, and shopping facility density, on the intensity of online car-hailing trips are more prominent, and the positive effect of factors such as population density, land-use mix, transportation facility density, road density, educational research institution density, and scenic spot density are relatively weak.

At the scale of the 1000 m grid, the built environment factors which have stronger positive effects on the online car-hailing trip intensity are the same as factors in the scale of grid 500 m, besides the density of living service facility density and population density, while the positive effects of land-use mix are relatively weak. The negative effects of medical facility density and sports and leisure facility density on the intensity of online car-hailing trips are stronger, while the negative effects of catering facility density are less significant. From this, it can be speculated that in a relatively fine scale, the effects of built environment factors such as density, land-use type, and distance to transit on the intensity of online car-hailing trip are significant. Our result is consistent with the results of previous studies proposed by Wang, Bi and Li et al. [6,7,31,37].

At the scale of the 1500 m grid, factors such as residential district density, accommodation service facility density, public service facility density, government agency density, shopping facility density, and population density have a greater positive effect on travel intensity, while bus stop density, sports and leisure facility density, and scenic spot density have a greater negative effect on travel intensity. Similar to the characteristics in the 1500 m grid, residential district density and shopping facility density are no longer factors with a greater positive effect on travel intensity at the scale of the 2000 m grid, while factors with a greater negative effect remain unchanged.

At the scale of the 2500 m grid, the factors such as public service facility density, accommodation service facility density, population density, and residential district density have a greater positive effect on travel intensity, while the factors that have a greater negative effect on travel intensity are still bus stop density and scenic spot density.

The characteristics of the 3000 m grid are more similar to those of the 2500 m grid. The difference is that the government agency density replaces the residential district density among the factors that have a greater positive effect on travel intensity, while the educational, scientific, and cultural facility density is included in the factors that have a greater negative effect on travel intensity.

At the 3500 m grid, the top three factors that have a positive effect on travel intensity are road density, transportation facility density, and shopping facility density, while the top two factors with a negative effect are government agency density and bus stop density. At the 4000 m grid, the top two factors that have a positive effect on travel intensity are government agency density and land-use mix, while the top three factors with a negative effect are residential district density, financial facility density, and accommodation service facility density.

At the scale of the 4500 m grid, two indicators, land-use mix and bus stop density, have a significant positive effect on the intensity of online car-hailing travel. At the scale of the 5000 m grid, the land-use mix is the only indicator that has a significant positive effect on the travel intensity.

Therefore, it can be summarized that the five types of built environment features have different impacts on online car-hailing travel intensity. As the scale of the analysis unit increases, the number of built environment factors that have a significant effect on travel intensity decreases. The differences in the effects of the five built environment factors on online car travel in different scales are described in detail below.

In the range of the 500 m grid to the 3000 m grid, density always has a positive effect on the intensity of online car-hailing trips. In the remaining analysis scales, density does not reflect an effect on the travel intensity. For this reason, it can be hypothesized that higher population densities can generate a higher intensity of online car trips. However, the prompting effect of population density can only be observed at small and medium scale of analysis. This result is inconsistent with the results of previous studies. Two studies investigated by Li Ting [7] and Wang Sai [6] showed no significant effect of population density on the traffic volume of online car-hailing. The reason may be that the population data used in these papers are the sixth census data and the study area and analysis unit of them are not consistent with our paper.

In the range of the 500 m grid to the 3500 m grid, there is a significant negative effect of public transportation proximity on the intensity of online car-hailing trips. On the contrary, there is a positive effect of public transportation proximity on the intensity of online car-hailing trips at the 4500 m grid. Therefore, it can be concluded that the effect of public transportation proximity on the intensity of online car-hailing trips shifts from a significant inhibitory effect to a facilitative effect as the scale of analyzed unit increases. The reason may be that there is a competitive relationship between public transportation and online car trips. When the travel distance is relatively short (maybe have a threshold), residents who live in the cells with well-developed public transportation are more likely to travel by public transportation than by online car-haling. When the travel distance exceeds the threshold, even in the cells with well-developed public transportation, residents are more likely to take online car-hailing to travel. This result also reflects that in long-distance travel behavior, the online car-hailing is more favored by the urban residents than public transportation.

In the range of the 500 m grid to the 1000 m grid, there is a slight positive effect of land-use mix on the intensity of online car-hailing trips. In the range of the 3500 m grid to the 5000 m grid, the land-use mix has a significant positive effect on the intensity of online car-hailing trips, and the effect force keeps increasing. In the range of the 1500 m grid to the 3000 m grid, the effect of land-use mix on the online car-hailing trip intensity is not significant. It is hypothesized that with the increasement of analysis scale, land-use mix gradually becomes the dominant factor influencing the intensity of online car-hailing trips. Therefore, a reasonable speculation is that enhancing the mixture of urban functional space can effectively promote the intensity of online car-hailing trips in the coarse spatial scales. Our finding is consistent with the findings of Li Ting et al. [7] and Cervero [10], but inconsistent with the findings of Munishi [57], Xie Weihan [18], Wang Sicheng [31], and Wang Sai [6]. The reason may be due to inconsistencies of the analysis scale and study area.

At the 500 m grid and 3500 m grid, the neighborhood design represented by road density has a positive effect on the intensity of online car-hailing trips but does not show a significant effect at the rest of the grid scales. It can be inferenced that the positive effect of road density on the intensity of online car-hailing trips can only be captured at certain scales. Our finding is inconsistent with studies investigated by Li Ting et al. [7]. Since the road density was excluded by multicollinearity analysis through VIF values, the authors concluded that road density had no significant effect on online car-hailing trip volume.

In the range of the 500 m grid to the 4000 m grid, the effects of land-use types on the intensity of online car-hailing travel were all significant and showed different characteristics with scale changes. Among them, the factors include public service facility density, residential district density, accommodation service facility density, and government agency density, showing significant positive effects at most of the analysis scales, while sports and leisure facility density and scenic spot density show significant negative effects. Our findings are consistent with previous studies.

It can be seen from Table 4 that the residual of the OLS model exhibits significant spatial autocorrelation in almost all grid scales. Unlike traditional cross-sectional data and panel data, the spatial correlation of the model residual will lead to spatial nonstationarity of the relationship. Compared with the ordinary square least regression model, spatial regression models such as the GWR model and the MGWR model can deal with the spatial nonstationarity more appropriately [58,59]. Therefore, the local regression models considering the spatial nonstationarity should be used for further analysis.

### 4.2. The Local Regression Results in Different Scales

The evaluation results of local regression models are summarized in Table 5.

By comparing the evaluation results of the OLS model (see Table 4) and the GWR model, it can be found that the GWR model is able to deal with spatial nonstationarity better than the OLS model at almost all grid scales. However, the GWR model residual distributions still exhibit some positive spatial autocorrelation at the 2000 m grid and the 2500 m grid. This result indicates that the GWR model cannot deal with spatial nonstationarity at all scales. Compared with the GWR model, the MGWR model fits better than the GWR model at almost all scales (evaluated by the AICc and the Moran’s I test results of the model residuals), except for the 4000 m grid. The possible reason is that the differences in the impact bandwidth of each factor are not significant under the 4000 m grid, so the results of the GWR model that use the average bandwidth are instead slightly better than the result of the MGWR model. The results demonstrate that the MGWR model can deal with the spatial nonstationarity problem better than the GWR model in the vast majority of scenarios. Therefore, the fitting results of the MGWR model will be used in the subsequent analysis of the estimated coefficient distribution patterns.

### 4.3. The Spatial Variation in Coefficient Estimation by MGWR Model in Different Scales

The estimated coefficient maps derived by the MGWR model in ten scales are shown Figure 2, Figure 3, Figure 4, Figure 5, Figure 6, Figure 7, Figure 8, Figure 9, Figure 10 and Figure 11.

To simplify the description, the analysis results of three scales, 500 m, 2500 m, and 5000 m, were selected as representatives of fine, medium, and coarse scales for detailed interpretation.

At the finest scale (the 500 m grid in our study), the effect of residential service density (including residential districts and accommodation service facilities) on the intensity of online car-hailing trips was mainly in the northeast of the study area. Among them, the density of accommodation service facilities generated a high-value cell aggregation area near the geometric center of the study area. This means that the contribution of residential density is biased over space in the study area. Our finding is consistent with the conclusion by Li et al. and Yang et al. that residential density would contribute to taxi trips positively [7,12].

The effect of financial facility density was also mainly presented in the northeastern part of the study area and generates two high-value cell aggregation areas. The high-value cells of the estimated coefficient of public service facility density were mainly distributed near the geometric center of the study area. This result is consistent with the actual situation in the study area. For example, Chunxi Road, which is close to the geometric center of the study area, is one of the prominent commercial areas in the study area. The Chunxi Road commercial area was well served by various public services facilities, while generating a high intensity of online car-hailing trips.

The estimated coefficient values of shopping facility density generated a high-value aggregation near the geometric center of the study area and showed a circle structure that gradually spreads from the inside to the outside. This result is consistent with the distribution of shopping facility POIs, which can also be demonstrated by the high shopping facility density in the Chunxi Road commercial area. The estimated coefficient values of the density of educational and research institutions gradually decreased from northeast to southwest, showing a wave-like pattern. The estimated coefficient values for the density of scenic spots were relatively small and showed a trend of decreasing from northwest to southeast. Similar to the distribution of the estimated coefficients of educational, research, and cultural institutions, the estimated coefficient values of the density of transportation facilities showed a wavy pattern that gradually decreases from northeast to southwest.

The density of bus stops showed a wavy pattern, decreasing gradually from north to south. The estimated coefficient values of population density were small overall. The coefficient values gradually decreased from northwest to southeast, indicating that its effect on the volume of online car-hailing shifts from positive to negative. The estimated coefficient values of land-use mix showed a decreasing pattern from west to east. The results indicate that the influence of land-use mix on the intensity of online car-hailing trips is mainly concentrated in the western region. As early as 1996, Randall Crane inferenced that the improved accessibility to multiple destinations increases nonwork taxi trips due to low trip costs [60], which is also demonstrated by our results. However, the impact pattern can only be observed in the coarse scales. The estimated coefficient values of road density also generated an inside-out decreasing circle pattern near the geometric center of the study area, indicating that the influence of road density is mainly limited to the region that is close to the geometric center of the study area. In summary, except for population density, land-use mix, and scenic spot density, the effects of the built environment factors on online car-hailing travel intensity showed a clear spatial variety in the 500 m grid.

At the medium scale (the 2500 m grid in our paper), the estimated coefficient values of public facility density generated a high-value aggregation near the geometric center of the study area and showed a circle structure that gradually spreads from the inside to the outside. The estimated coefficients of accommodation service density showed a decreasing pattern from northwest to southeast, which indicates that the influence of accommodation service density on the intensity of online car-hailing trips at this scale is mainly concentrated in the northwestern region. The estimated coefficient values of scenic spot density were smaller and showed a decreasing trend from north to south. The estimated coefficient values of the density of accommodation service facilities were all negative and showed a decreasing trend from north to south. The estimated coefficient values of population density showed a pattern of decreasing from east to west, but the spatial variation was not significant. The density of bus stops generated a high-value aggregation near the geometric center of the study area, and the spatial variation in the overall pattern was more significant. In conclusion, except for the density of accommodation services, population density, and scenic spot density, the selected built environment factors both showed significant spatial variety in their effect on online car-hailing travel intensity.

At the coarsest scale (the 5000 m grid in our study), the land-use mix is the only factor that has a significant effect on the intensity of online car-hailing travel. The estimated coefficient values of land-use mix gradually decreased from north to south, showing a wave-like pattern. At the remaining scales, the effects of built environment factors on the online car-hailing trip intensity showed different spatial variation characteristics.

In conclusion, the effect of built environment factors on the intensity of online car-hailing trips showed different spatial characteristics at different scales. At ten scales involved in our study, the effect of population density on the intensity of online car-hailing trips showed a spatial pattern of decreasing from north to south. At the finest scale (the 500 m grid in our study), the effect of road density on the travel intensity showed a circle pattern spreading from inside to outside. At the coarser scale (the 3500 m grid in our study), the effect of road density on the intensity of online car-hailing trips showed a wave-like pattern decreasing from southeast to northwest. At the finer scale (the 500 m and 1000 m grids in our study), the spatial variation in the effect of land-use mix on the online car-hailing travel intensity was not significant. At the coarser scale (the 4500 m and 5000 m grids in our study), the spatial variation in the effect of land-use mix on the intensity of online car-hailing trips can be clearly identified. Accordingly, a reasonable inference is that the land-use mix may have a larger scale of effect on the intensity of online car-hailing travel.

At the finest scale (the 500 m grid in this paper), the effect of bus stop density on the intensity of online car-hailing trips showed a wave-like pattern with significant spatial variation. At the medium scale (the range from the 1500 m grid to 3000 m grid in this paper), the effect of bus stop density on the intensity of online car-hailing trips showed a pattern of higher inside and lower outside, with significant spatial variation. At the coarser scales (the 3500 m grid and 4500 m grid in this paper), the spatial variation in the effect of bus stop density was not significant. This result demonstrated the competitive relationship between public transportation proximity and online car-hailing travel in short distance trips. The land-use type had both positive and negative effects at different scales and showed different spatial patterns. A common feature was that the spatial variability of the effect of various land-use type factors becomes smaller as the scale increases.

## 5. Conclusions and Prospect

This paper analyzed the effects of urban built environment factors on the intensity of online car-hailing trips by applying the OLS model, the GWR model, and the MGWR model in ten grid scales of Chengdu, China. The main findings can be summarized as follows.

(1)The built environment factors that significantly affect the intensity of the online car-hailing travel varied under different grid scales. As the grid size increases, the number of built environment factors that have significant effects on trip intensity decreased continuously.(2)From the 500 m grid to the 3000 m grid, there was always a positive effect of population density on the online car-hailing travel intensity. As the analysis scale increased, the effect of proximity to public transportation on the online car-hailing travel intensity shifted from an obvious inhibitory effect to a certain degree of facilitation. Meanwhile, the positive effect of land-use mix on the online car-hailing travel intensity was more and more significant. At the 500 m grid and 3500 m grid, road density had a positive effect on the online car-hailing travel intensity, but its effect was not significant at the remaining analysis scales. From the 500 m grid to the 4000 m grid, land-use type had both positive and negative effects on the online car-hailing travel intensity and showed different characteristics at different scales.(3)The results of the Moran’s I test for the OLS model residual indicated that there is spatial nonstationary in the effect of the built environment on the online car-hailing traveling intensity in the study area. By comparing the fitting results of involved models, it can be found that both the GWR model and the MGWR model can cope with spatial nonstationary better than the OLS model at almost all scales, and the performance of the MGWR model is better than that of the GWR model.(4)At different scales, the effects of built environment factors on the online car-hailing trip intensity showed different spatial variability characteristics. At various scales, the effect of population density on the online car-hailing trip intensity gradually decreased from north to south. The spatial patterns of the effect of road network density on the online car-hailing trip intensity include circling and wave patterns, in that the former is a characteristic at relatively fine scales while the latter at relatively coarse scales. The spatial variation in the effect of land-use mix on the online car-hailing trip intensity can only be identified more significantly at a relatively coarse scale. At the relatively fine and medium scales, the effect of bus stop density on the online car-hailing traveling intensity showed a wave-like pattern and a circle-like pattern. At the relatively coarse scale, the spatial variation in its effect was not obvious. The effect of various land-use types showed different spatial patterns at different scales, including wave-like pattern, circle-like pattern, and multi-core-like pattern. The spatial variation in the effects of various land-use factors gradually decreased with the increase in the analysis scale. It can be inferenced that different analysis scales may lead to different and even contrary conclusions.

In terms of theoretical contributions, the variation in the influence of the built environment on the intensity of online car-hailing trips at different scales obtained in this study provides some impetus for solving the MAUP problem in the online car-hailing travel behavior study. In terms of practical value, the findings of this paper can not only help policy makers to better understand the spatiotemporal characteristics of online car-hailing travel behavior in the city of Chengdu, but also can help to develop more refined transportation optimization policies, especially with the consideration of variation in the effect among the analysis scales.

Undoubtfully, our study still has some limitations. For instance, urban land use is represented using the POI data, which are not detailed enough to capture the land use accurately. Therefore, more research on fine-scale land-use data is needed in the future. In addition, the traffic analysis unit in this paper has only one type of grid, and the criteria for dividing the grid size are not fine-grained enough, thus potentially making the results not comprehensive. In the future, more refined analysis unit types and sizes should be incorporated to comprehensively explore the modifiable analysis unit problem issue in the study on online car-hailing traveling and the built environment. Finally, the study area is only a local area of Chengdu, which may lead to different and even conflicting conclusions in different areas. Therefore, more empirical research is needed to validate whether our findings hold in other cities.

## Figures and Tables

**Figure 1 ijerph-19-05325-f001:**
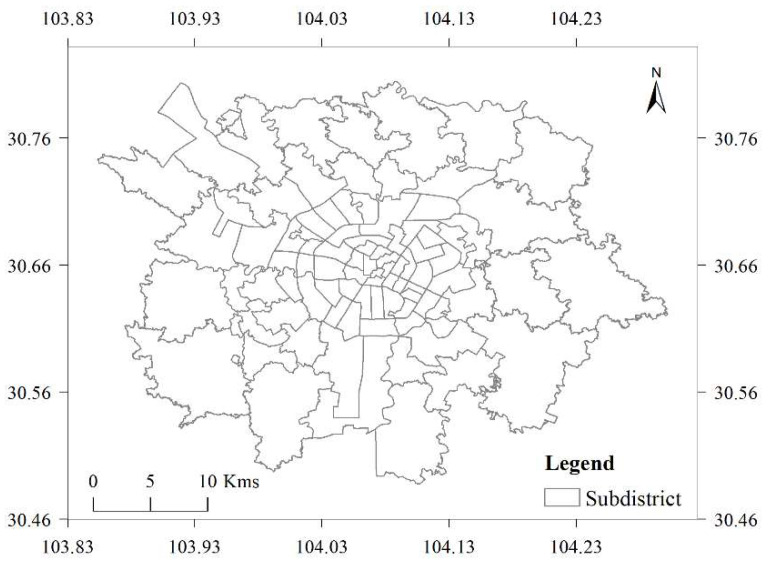
The study area map.

**Figure 2 ijerph-19-05325-f002:**
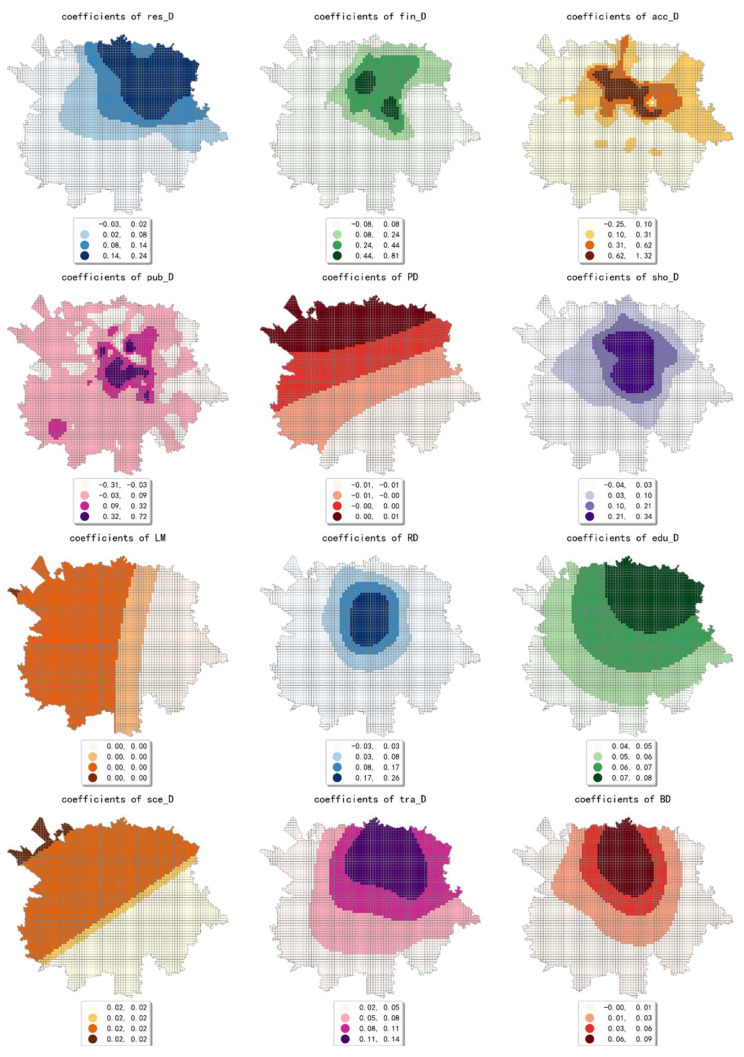
The estimated coefficient maps in the grid with 500 m.

**Figure 3 ijerph-19-05325-f003:**
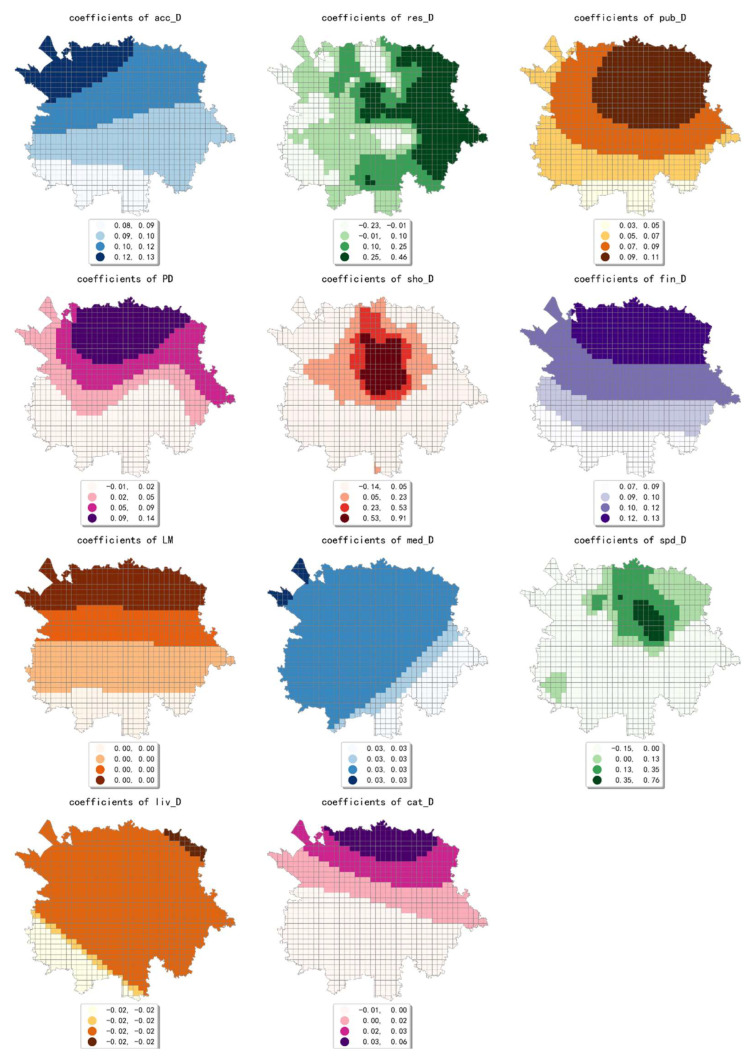
The estimated coefficient maps in the grid with 1000 m.

**Figure 4 ijerph-19-05325-f004:**
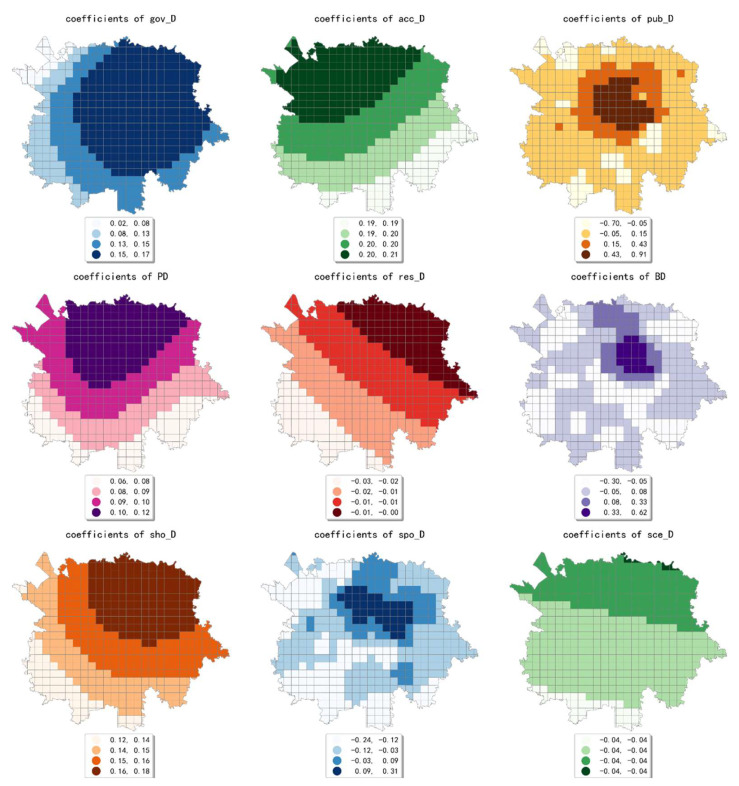
The estimated coefficient maps in the grid with 1500 m.

**Figure 5 ijerph-19-05325-f005:**
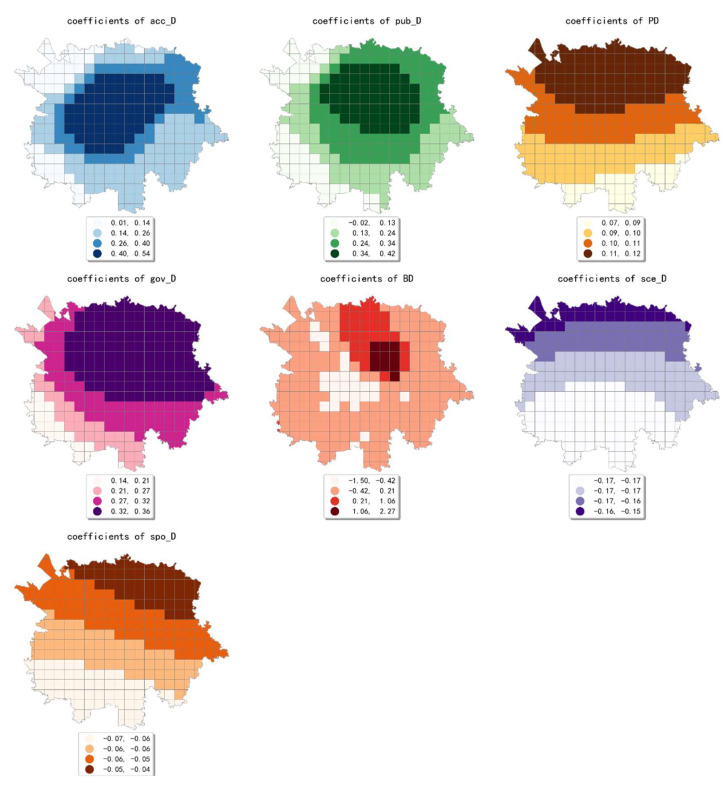
The estimated coefficient maps in the grid with 2000 m.

**Figure 6 ijerph-19-05325-f006:**
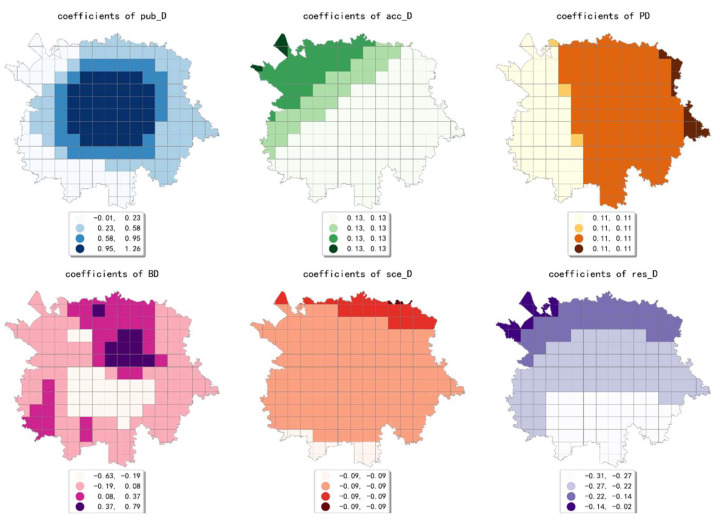
The estimated coefficient maps in the grid with 2500 m.

**Figure 7 ijerph-19-05325-f007:**
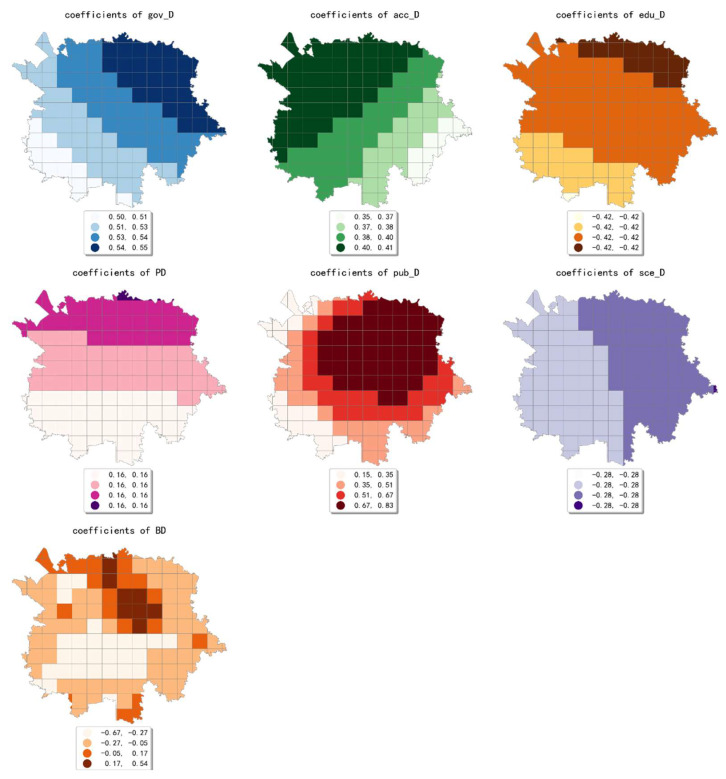
The estimated coefficient maps in the grid with 3000 m.

**Figure 8 ijerph-19-05325-f008:**
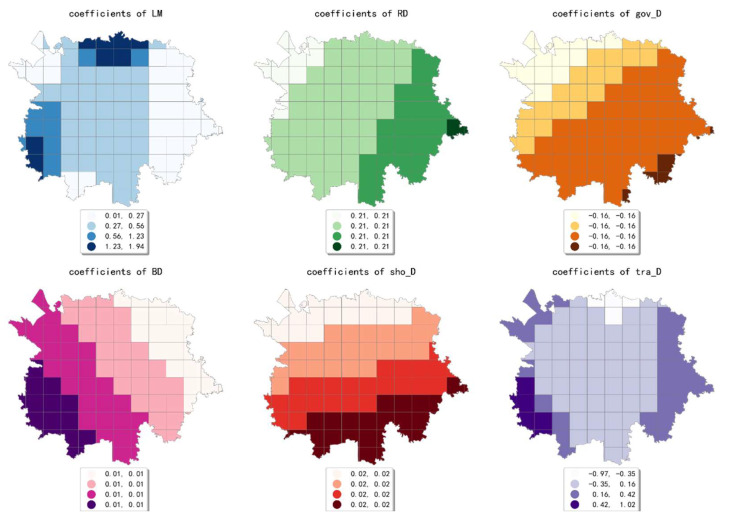
The estimated coefficient maps in the grid with 3500 m.

**Figure 9 ijerph-19-05325-f009:**
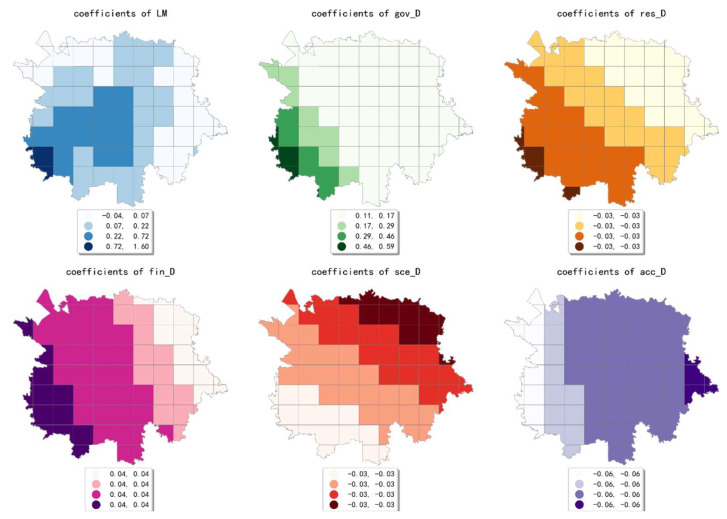
The estimated coefficient maps in the grid with 4000 m.

**Figure 10 ijerph-19-05325-f010:**
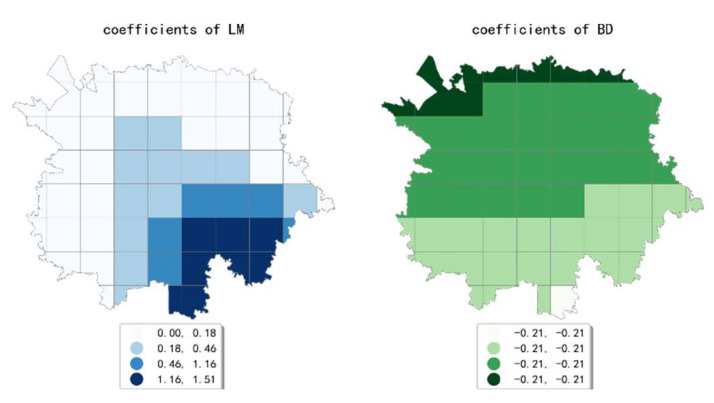
The estimated coefficients map in the grid with 4500 m.

**Figure 11 ijerph-19-05325-f011:**
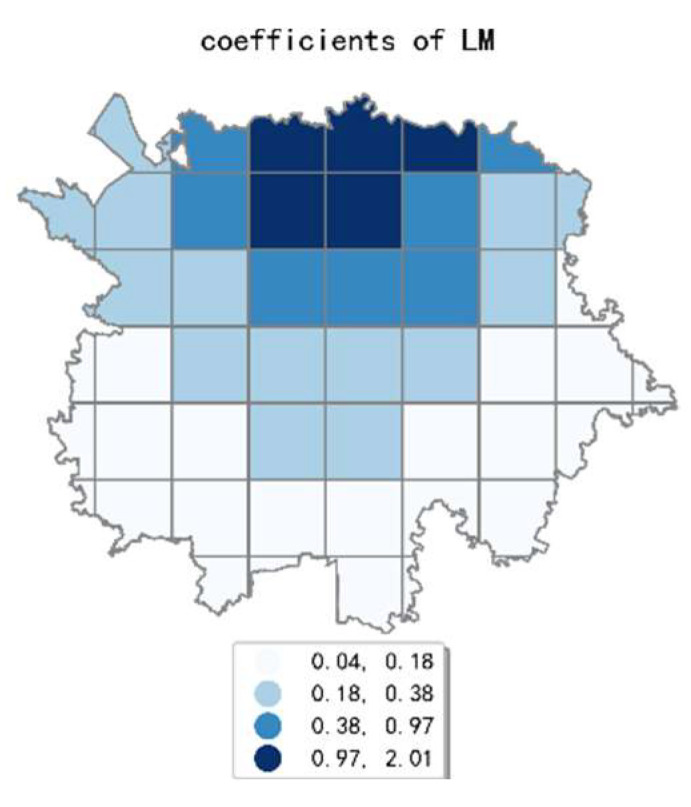
The estimated coefficient maps in the grid with 5000 m.

**Table 1 ijerph-19-05325-t001:** The indicators of urban built environment based on “five Ds” principle.

Features	Main Indicators in Previous Literatures	Indicators in Our Study
density	population density, employment density	population density (PD)
diversity	land-use mix, job–housing imbalance	land-use mix (LM)
design	neighborhood, road density, etc.	road density (RD)
distance to transit	bus stop density, distance to metro, etc.	bus stop density (BD)
different land-use types	−	POI density

**Table 2 ijerph-19-05325-t002:** The abbreviations of indicators.

Indicators in Our Study	Abbreviations
population density	PD
land-use mix	LM
road density	RD
bus stop density	BD
catering facility density	cat_D
scenic spot density	sce_D
public service facility density	pub_D
company density	com_D
shopping facility density	sho_D
transportation facility density	tra_D
financial facility density	fin_D
educational, scientific, and cultural facility density	edu_D
residential district density	res_D
living service facility density	liv_D
sports and leisure facility density	spo_D
medical service facility density	med_D
government agency density	gov_D
accommodation service facility density	acc_D

**Table 3 ijerph-19-05325-t003:** The VIF values of initial variables in ten scales.

Variables					VIF Values in Different Scales					
	500 m	1000 m	1500 m	2000 m	2500 m	3000 m	3500 m	4000 m	4500 m	5000 m
PD	1.31	1.48	1.96	2.22	2.40	3.10	3.68	6.90	1.28	6.43
RD	1.49	1.93	1.96	1.78	2.18	1.28	5.87	5.65	2810.33	2.76
BD	1.74	−	4.14	5.36	3.53	4.85	10.41	9.21	2773.74	−
LM	1.65	1.44	1.38	1.33	1.31	1.65	2.02	1.80	2.29	1.61
acc_D	1.45	2.20	2.62	4.28	5.10	6.54	7.59	8.85	20.04	17.24
cat_D	5.22	−	23.32	28.28	41.87	48.72	105.02	111.09	159.23	−
com_D	1.95	2.76	3.64	4.65	4.89	7.26	9.99	11.40	14.40	33.39
spo_D	3.27	6.75	10.51	17.64	26.85	33.58	66.73	79.37	82.84	152.48
gov_D	2.05	3.66	5.86	9.41	11.55	26.83	48.56	6.49	52.51	84.32
fin_D	3.30	6.35	11.45	18.21	17.67	28.63	58.06	50.37	71.78	49.36
liv_D	5.51	8.60	17.83	26.38	43.15	50.57	77.50	93.28	191.42	219.01
med_D	2.55	4.84	6.43	10.94	21.78	21.32	57.37	44.59	60.18	178.04
pub_D	1.69	2.81	4.58	7.19	11.18	16.65	31.21	25.10	23.59	48.20
res_D	3.85	5.86	7.74	13.56	21.97	36.93	52.25	29.12	92.19	68.57
sce_D	1.21	1.63	2.10	3.45	4.21	7.25	10.50	10.14	20.67	20.19
edu_D	3.29	7.01	9.79	19.10	29.67	46.26	95.77	82.53	102.28	180.11
sho_D	2.45	4.82	9.30	13.79	19.84	26.89	57.56	42.83	52.01	157.12
tra_D	5.19	10.80	17.76	27.95	31.89	45.85	82.30	105.11	113.41	110.55

**Table 4 ijerph-19-05325-t004:** The regression results of OLS model in ten scales.

Variables						Coefficients and SE in Different Scales				
	500 m	1000 m	1500 m	2000 m	2500 m	3000 m	3500 m	4000 m	4500 m	**5000 m**
PD	0.087 (0.012)	0.125 (0.021)	0.164 (0.031)	0.205 (0.038)	0.186 (0.047)	0.263 (0.055)	−	−	−	−
RD	0.053 (0.013)	−	−	−	−	−	0.997 (0.143)	−	−	−
BD	−0.034 (0.013)	−	−0.153 (0.040)	−0.183 (0.041)	−0.196 (0.051)	−0.212 (0.063)	−0.714 (0.179)	−	0.274 (0.097)	−
LM	0.075 (0.013)	0.040 (0.040)	−	−	−	−	0.286 (0.085)	0.125 (0.034)	0.569 (0.097)	0.578 (0.110)
acc_D	0.167 (0.012)	0.306 (0.025)	0.312 (0.036)	0.454 (0.052)	0.328 (0.060)	0.515 (0.081)	−	−0.176 (0.057)	−	−
cat_D	−	−	−	−	−	−	−	−	−	−
com_D	−	−	−	−	−	−	−	−	−	−
spo_D	−	−	−0.129 (0.057)	−0.195 (0.060)	−	−	−	−	−	−
gov_D	−	−	0.193 (0.050)	0.391 (0.064)	−	0.727 (0.123)	−1.192 (0.284)	1.903 (0.067)	−	−
fin_D	0.129 (0.018)	0.156 (0.035)	−	−	−	−	−	−0.375 (0.105)	−	−
liv_D	−	0.133 (0.051)	−	−	−	−	−	−	−	−
med_D	−	−0.125 (0.038)	−	−	−	−	−	−	−	−
pub_D	0.146 (0.013)	0.202 (0.027)	0.251 (0.048)	0.362 (0.067)	0.546 (0.098)	0.491 (0.109)	−	−	−	−
res_D	0.261 (0.017)	0.245 (0.034)	0.208 (0.055)	−	0.159 (0.069)	−	−	−1.035 (0.098)	−	−
sce_D	0.033 (0.011)	−	−0.070 (0.032)	−0.184 (0.045)	−0.135 (0.055)	−0.281 (0.073)	−	−0.174 (0.070)	−	−
edu_D	0.043 (0.017)	−	−	−	−	−0.613 (0.159)	−	−	−	−
sho_D	0.103 (0.014)	0.129 (0.038)	0.166 (0.050)	−	−	−	0.581 (0.261)	−	−	−
tra_D	0.075 (0.023)	−	−	−	−	−	0.764 (0.163)	−	−	−
**Metrics**	**500 m**	**1000 m**	**1500 m**	**2000 m**	**2500 m**	**3000 m**	**3500 m**	**4000 m**	**4500 m**	**5000 m**
Adj. R^2^	0.572	0.671	0.730	0.771	0.774	0.821	0.586	0.931	0.548	0.321
AIC	8053.725	1862.015	779.284	409.614	273.811	171.820	218.667	24.577	143.535	145.077
RSS	1723.607	348.318	132.847	65.224	42.660	24.317	41.410	5.204	29.374	37.985
Moran I of Residual	0.508	0.529	0.515	0.462	0.408	0.247	0.128	0.014	0.072	0.022
*p*-value	0.000	0.000	0.000	0.000	0.000	0.000	0.009	0.598	0.215	0.623

**Table 5 ijerph-19-05325-t005:** Model evaluation results of local models in different scales.

Scales	Models	Adj. R^2^	AIC	RSS	Bandwidth	Moran I of Residual	*p*-Value
500 m	GWR	0.922	2713.854	235.732	1143.620	−0.017	0.033
	MGWR	0.917	2245.675	281.408	Varied ^1^	−0.034	0.000
1000 m	GWR	0.946	515.039	39.678	1924.210	0.025	0.110
	MGWR	0.967	−140.070	26.100	Varied	−0.075	0.000
1500 m	GWR	0.955	207.513	14.360	2466.280	0.061	0.010
	MGWR	0.969	173.983	8.683	Varied	−0.044	0.101
2000 m	GWR	0.942	183.366	11.281	3066.800	0.110	0.000
	MGWR	0.972	99.093	4.477	Varied	−0.021	0.590
2500 m	GWR	0.935	145.831	8.402	3524.190	0.131	0.000
	MGWR	0.967	111.092	3.389	Varied	−0.059	0.183
3000 m	GWR	0.923	109.618	7.868	5290.090	0.028	0.445
	MGWR	0.957	39.289	2.941	Varied	−0.080	0.139
3500 m	GWR	0.944	49.544	4.202	5862.660	0.013	0.677
	MGWR	0.957	39.289	2.941	Varied	−0.009	0.999
4000 m	GWR	0.982	−32.981	0.928	5511.2	−0.026	0.813
	MGWR	0.970	41.006	1.305	Varied	−0.047	0.587
4500 m	GWR	0.980	−39.450	0.961	5414.45	−0.132	0.032
	MGWR	0.978	−30.664	1.085	Varied	−0.001	0.834
5000 m	GWR	0.973	−16.039	1.103	5223.95	−0.099	0.173
	MGWR	0.981	−20.242	0.712	Varied	−0.002	0.850

^1^ The bandwidth of factors in the MGWR model are varied, not fixed.

## Data Availability

The data used to support this study are provided by the Didi Chuxing GAIA Initiative. Due to the security policies, the authors have no right to disclose, publish, copy, or conduct all original datasets. Readers in need, please use the school/research institution email to contact gaia@didiglobal.com for further information.

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
