# Peer review of "How Does the Urban Built Environment Affect Online Car-Hailing Ridership Intensity among Different Scales?"

_ijerph, 2022, doi:10.3390/ijerph19095325_

Round 1

Reviewer 1 Report

This article looks like a learning note of statistics rather than an academic article.

They analyze with OLS, GWR and MGWR. However, for OLS, in lines 299-300 they say "cannot reflect the true spatial characteristics." That means OLS is an inappropriate analytical tool for their research. So why do they give OLS results? Besides, in section 3.3.3 they discuss OLS regression like a statistics textbook, which is also inappropriate for the description of an academic article.

As for the way the authors show results, it is difficult to get implications. For example, in table 3, they give a list of variables surviving the step wise procedure, and in table 4 they give coefficients of each variable. I wondered what the meaning of giving table 3 is when table 4 follows immediately. And in table 4, why don't they give standard deviations for each variable, not only coefficients? They should give the statistical significance of each independent variable. Without doing that, we don't know which variables are actually effective and how much. And they have to give some explanations why some variables are chosen by stepwise method and others are not, using expert knowledge such as social context not only for "computer recommendations".

If the authors have an appropriate research question, they are forced to choose appropriate scales for their analysis. Without hypothesis, they leave all of their choice of variables to the computer. Lack of appropriate hypothesis leads to their current approach!

There are a lot of analytical failures in the paper. To take an example, line 434-437, if they want to detect the effect of public transportation, they argue that grids need to be smaller than 4500m. But I wonder if they can analyze with 500m scale, why they choose larger scale. Basically, they must use a 500m scale, and if they want to know about anything of larger scale, they should consider the firstly the relationship between consecutive grids. In so doing, they can argue why some variables are chosen by the stepwise method.

There are many other failures to be addressed in this article. But I already give enough reasons for the judgement.

The authors have just learnt how to use analytical tools and have been at the start point of writing an article. They must look for appropriate research questions from now.

Therefore, I am of the opinion that this article is NOT qualified for publication from the ijrph.

Reviewer 2 Report

Improve the quality of figures. The annotations in Figure 1 make it hard to read. The author should mark the actual study area since the main text says, “study area was set to coincide with the area where the 89 streets are located.” Besides, from Figure 2 to Figure 11, the mesh lines are too thick and suppress the more critical information.

As mentioned in the paper, “the urban built environment can be measured from five aspects including design, diversity, density, distance to transit, and destination accessibility.” Then why do the authors use design, diversity, density, distance to transit, and different land use types instead?

Check the equations, the formula of MGWR is wrong.

Since there are five different types of built environment features. I wonder how they (say POI density in total) vary with scale increase.

From line 434 to line 436, the authors state, “When the travel distance is within a certain threshold (e.g., 4500m in this paper), residents lived in the cells with well-developed public transportation are more likely to travel by public transportation than by online car-haling.” The analytical unit is 4500m doesn’t necessarily mean that the travel distance is also 4500m.

Line 447, Therefore, a reasonable speculation is that enhancing the mixture of urban functional space can effectively promote the intensity of online car-hailing trips in the coarse spatial scales. The intensity of online car-hailing is the intensity of what? The pick-ups or the drop-offs? It’s natural to think that if there is a good mixture of urban functions, the space is self-provided so that people don’t need to travel away to get their requirements, and people will travel here to satisfy their demands.

Reviewer 3 Report

In this text, the authors provide a multiscale exploratory study conducted in Chengdu, China, using the step-wise regression selection and three spatial regression models to evaluate the effects of factors and study scales on online car-hailing ridership. They state that their analysis had three main findings regarding the size of spatial grids and environmental factors as relevant to the results of analyses.

The central positive aspect of this text is the scientific soundness of the statistical modeling and analyses. The authors provide a set of three different evaluations to answer how urban environment factors affect car-hailing ridership intensity. This three-part research provides an excellent understanding of the thought process.

The most damaging aspect is the difficulty of understanding the research question and some aspects of the text. The abstract does not provide a clue on the main research question, which is only established at the end of the introductory section. The abstract does not help to identify the question. Furthermore, it is hard to follow the thought process at the beginning of the text due to long and disorganized paragraphs and ideas. Finally, the results are only understandable when reading the conclusion.

All information is there, but the text organization and presentation require more work before publishing.

Further comments about style are

  • The authors often (or mostly) did not put spaces between words and citation brackets. This should be corrected.
  • Figure 1 is overloaded with information. The authors could remove the texts, indicating a range of geographical coordinates. Alternatively, they can change the scale of the fonts and enlarge the size of the map.
  • Often, the authors use long paragraphs presenting several topics. This style of writing does not help the readers in understanding the results. They should reorganize their paragraphs, dividing them according to the topics they contribute to.
  • Moderate proofreading is required to improve the text quality.
  • The abstract needs to be rewritten. The authors spent much effort explaining details of intermediate steps in the methodology, while the research question was unclear, and no conclusions were presented.

Round 2

Reviewer 1 Report

I now understand what the authors try to make clear and their research question after reading comments on the discussion of the MAUP. And in the revised manuscript, many of the things I wondered about are properly explained.

#Suggestions#

In lines 287-297, I recommend the authors summarize in a tabular form. Many "short for" make it difficult to follow "abbreviation of what."

As for tables 3 and 4, I recommend rewriting the tables as in the attached file. That is because the main contribution of the paper is to show which explanatory variables correlate with car-hailing and others are not according to different scales. In my way of drawing a table, readers can easily follow the pattern of how variables are discarded as the chosen scale sizes increase at a glance of the combination of filled and empty cells.

As for the OLS model, the authors say in Line 334-335, "which cannot reflect the true spatial characteristics," and in the authors' comment, "the OLS model is classified as a global regression model, the GWR series model is classified as a local regression model." I prefer the latter expression because the former sounds a little negative toward the OLS, which is also important as it is actually used to identify the significant built environment variables using AIC.

I recommend moving figures 2-11 to the appendix, as the authors say in lines 556-558, "selected as representatives," which means they are enough to understand the implications. I would put 500m figures around between Line 559-601, 2500m figures Line 602 -617, and so on, for easy reference. And then, more interested readers can move to the appendix.

The above comments are my recommendations, but the authors don't necessarily follow them.

Otherwise is OK
